# 5-Fluorouracil-Immobilized Hyaluronic Acid Hydrogel Arrays on an Electrospun Bilayer Membrane as a Drug Patch

**DOI:** 10.3390/bioengineering9120742

**Published:** 2022-11-30

**Authors:** Ji-Eun Lee, Seung-Min Lee, Chang-Beom Kim, Kwang-Ho Lee

**Affiliations:** 1Graduate Program of Advanced Functional Materials and Devices Development, College of Engineering, Kangwon National University, Kangwondaehak-gil, Chuncheon-si 24341, Kangwon-do, Republic of Korea; 2Interdisciplinary Program in Advanced Functional Materials and Devices Development, College of Engineering, Kangwon National University, Kangwondaehak-gil, Chuncheon-si 24341, Kangwon-do, Republic of Korea; 3Intelligent Robot Research Team, Electronics and Telecommunications Research Institute, Daejeon-si 34129, Chungcheongnam-do, Republic of Korea; 4Division of Mechanical and Biomedical, Mechatronics and Materials Science and Engineering, College of Art, Culture and Engineering, Kangwon National University, Kangwondaehak-gil, Chuncheon-si 24341, Kangwon-do, Republic of Korea

**Keywords:** 5-fluorouracil, drug patch, hyaluronic acid, electrospun membrane, hydrogel array

## Abstract

The hyaluronic acid (HA) hydrogel array was employed for immobilization of 5-fluorouracil (5-FU), and the electrospun bilayer (hydrophilic: polyurethane/pluronic F-127 and hydrophobic: polyurethane) membrane was used to support the HA hydrogel array as a patch. To visualize the drug propagating phenomenon into tissues, we experimentally investigated how FITC-BSA diffused into the tissue by applying hydrogel patches to porcine tissue samples. The diffusive phenomenon basically depends on the FITC-BSA diffusion coefficient in the hydrogel, and the degree of diffusion of FITC-BSA may be affected by the concentration of HA hydrogel, which demonstrates that the high density of HA hydrogel inhibits the diffusive FITC-BSA migration toward the low concentration region. YD-10B cells were employed to investigate the release of 5-FU from the HA array on the bilayer membrane. In the control group, YD-10B cell viability was over 98% after 3 days. However, in the 5-FU-immobilized HA hydrogel array, most of the YD-10B cells were not attached to the bilayer membrane used as a scaffold. These results suggest that 5-FU was locally released and initiated the death of the YD-10B cells. Our results show that 5-FU immobilized on HA arrays significantly reduces YD-10B cell adhesion and proliferation, affecting cells even early in the cell culture. Our results suggest that when 5-FU is immobilized in the HA hydrogel array on the bilayer membrane as a drug patch, it is possible to control the drug concentration, to release it continuously, and that the patch can be applied locally to the targeted tumor site and administer the drug in a time-stable manner. Therefore, the developed bilayer membrane-based HA hydrogel array patch can be considered for sustained release of the drug in biomedical applications.

## 1. Introduction

Cancer, one of the leading causes of death worldwide, is a disease wherein abnormal cells grow uncontrollably and form a mass called a tumor [1]. Tumor cells can sometimes migrate to surrounding host tissues through blood circulation or lymph nodes; this process is commonly referred to as metastasis. The main treatments for cancer are surgery, radiation therapy, and chemotherapy. In brief, surgery is a common treatment for many types of cancer, wherein the tumor mass and some of the nearby tissues are excised [2]. Radiation therapy uses high-energy beams to destroy cancer cells and shrink tumors [3]. Chemotherapy is a systemic treatment that uses anticancer drugs to prevent or kill cancer cells based on injections and orally administered pills [4,5,6]. Usually, anticancer drugs introduced into the body are transported through blood vessels and circulate throughout the body to prevent the growth of or kill cancer cells [7,8,9]. Despite the convenience of use and high effectiveness, most anticancer drugs are cytotoxic and kill cancer cells (that rapidly divide) by blocking division; this leads to side effects that damage normal cells [10,11,12,13]. Molecularly targeted therapies are becoming a preferred alternative in that they can minimize the side effects of anticancer drugs; however, personalized medicine (targeting individual patients) has a high cost. Recently, to increase the effects of a drug and minimize side effects according to a patient’s disease or physical characteristics, drug delivery systems have been proposed to efficiently deliver a drug to a target site [14,15,16,17,18,19]. It is an essential technology used to selectively deliver a drug according to the site of the disease or to release the drug periodically while controlling the released amount. However, the effectiveness of these systems is still not satisfactory for clinical applications [20,21,22,23,24,25,26]. Several studies have been conducted to create micro- or nanosized ultrafine structures, based on micro- and nanotechnologies, to deliver cancer-killing drugs to cancer cells [27,28,29,30,31,32,33,34,35,36]. Recently, the microneedle presented a new drug delivery method that overcame the limitations of existing drug delivery methods and had pain relief and rapid drug delivery through minimal penetration of skin [37,38,39]. Using various materials, numerous drug delivery carriers have been developed, including polymers, nanolipids, and inorganic substances; however, drug-encapsulated carriers are small and have low renal clearance, meaning they avoid rapid excretion from the blood circulation. Another factor is that tumor microvessels with high interstitial fluid pressure have a high intratumoral drug penetration rate, thereby reducing the therapeutic effect of the drug.

Hydrogels, which have been attracting attention recently, are being used for continuous administration of hydrophilic and hydrophobic biomolecules and can have a high drug-loading dose regardless of the microvascular system of the tumor. Accordingly, controlled release of the drug has become possible by controlling the fiber concentration [40,41,42,43,44,45,46,47,48]. Recently developed conjugated polymer-based hydrogels have been expanding the accessible materials for emerging drug delivery systems; they possess high biocompatibility and conductivity. The long-distance and long-term stable operation required that polymer-based hydrogels used as a drug carrier were on a micro or macro scale. Recently, manufacturing techniques, such as electrochemical patterning and screen printing, have been proposed to synthesize a thermal, photo, and pH-responsive hydrogel. However, low resolution in two dimensions, high cost, and mask films are still problems to be overcome [49,50]. In particular, when a hydrogel and multiple drugs are administered in combination, the anticancer effect of the drug increases, and drug tolerance decreases [51,52,53,54]. Whenever strong anticancer drugs are developed, their effective delivery to the cancer cell site with minimal side effects is always a challenge. Hyaluronic acid (HA), also known as hyaluronan, is a material found in the vitreous body and extracellular matrix (ECM) of cartilaginous tissue. HA is an essential factor of the ECM and affects cell signaling, the wound healing process, morphogenesis, and matrix organization. 5-fluorouracil (5-FU) is a standard chemotherapeutic agent and is extensively used in the treatment of several cancer types, such as gastric, colon, lung, breast, pancreatic, rectal, and head and neck cancers. 5-FU is metabolized to 5-fluorodeoxyuridylate, suppressing DNA synthesis by inhibiting thymidylate synthase. Additionally, the drug is also converted to 5-fluorouridine triphosphate, which is incorporated into RNA to inhibit RNA synthesis.

Herein, electrospun bilayer membranes, which have simultaneous properties (hydrophobic or hydrophilic) on each side, were used as a structural scaffold for local release patches. The 5-FU-immobilized HA hydrogel array was easily produced with the photo-crosslinking process. We characterized the repellency or absorption properties of the bilayer membrane and demonstrated a diffusion ratio aspect depending on HA concentrations. In addition, HA hydrogel arrays were investigated using practical tests with porcine tissue and analyzing the effect of 5-FU on the YD-10B cell line (oral cancer cells). The proposed carrier that modulates the diffusion of a drug by adhering at a specific site can be loaded with a drug to achieve its local release control. We believe that it will be beneficial in numerous applications, including chemotherapy and tissue regeneration.

## 2. Materials and Methods

### 2.1. Three-Dimensional (3D)-Printing-Based Master Micromold

The master mold, which has convex patterns to allow for the construction of the HA hydrogel array, was produced using PolyJet 3D printing. Vero (RGD 824, Stratasys Ltd., Eden Prairie, MN, USA) was used as a material for the master mold in conjunction with the use of the PolyJet 3D printer (J826 prime, Stratasys Ltd.). Vero offers excellent visualization with a tensile strength in the range of 60–70 MPa, a flexural strength in the range of 75–110 MPa, and a heat deflection temperature in the range of 45–50 °C. The Vero-based convex pattern on the master mold surface was designed as a hemisphere with a diameter of 700 µm, and the stacked layer thickness was set at 14 µm (Appendix A). After the printing of the master mold, PDMS (Dow Corning, Midland, MI, USA), the Si-based organic polymer with excellent optical transparency and mechanical properties, was used to replicate the concave patterns from the convex patterns on the Vero master mold. The PDMS solution was prepared by mixing the base polymer and the curing agent at the ratio of 10:1.5 wt %. The mixed PDMS solution was maintained in a vacuum chamber for 1 h to eliminate bubbles and was poured onto the Vero master mold. To prevent thermal deformation, the Vero master mold and PDMS solution were cured at 43 °C for 24 h. After detaching the Vero master mold, the replicated PDMS mold containing concave patterns for HA hydrogel arraying was obtained (Appendix A).

### 2.2. Synthesis of HA Hydrogel

HA, a nontoxic degradation product, is one of the most suitable biomaterials, which has superior biocompatibility characteristics [55,56,57,58,59,60,61,62]. To synthesize HA hydrogel, sodium hyaluronate (40 kDa, Lifecore, Chaska, MN, USA) was dissolved in 100 mL deionized (DI) water to a concentration of 1% (*w/v*), and 1 mL of methacrylic anhydride (MA) (Sigma-Aldrich, St. Louis, MO, USA) was added to the solution. The solution was adjusted to pH 8 following the slow addition of an aqueous solution of 5 N NaOH (Sigma-Aldrich) and continued to synthesize at 7 °C for 24 h to preserve reaction temperature. To remove unnecessary components, the solution was then dialyzed with DI water for at least 48 h using a dialysis bag (molecular weight cutoffs: 12–14 kDa, Spectrum Laboratories, Piscataway, NJ, USA). After dialysis, the solution was lyophilized using a freeze dryer (TFD, IlshinBioBase, Dongducheon-si, Republic of Korea) at −68 °C and 660 Pa for 4 days and was stored in a deep freezer at −70 °C in a powder state for future use. The synthesized HA hydrogel was verified using a Fourier transform infrared (FT-IR) spectrometer (iN10/iS50, Thermo Scientific, Waltham, MA, USA).

### 2.3. Fabrication of Electrospun Bilayer Membrane

To produce the electrospun bilayer membrane, a membrane with hydrophobic and hydrophilic surfaces was electrospun sequentially. First, the solution for hydrophobic fibers was prepared to dissolve 15% (*w/v*) of polyurethane (PU) (Dow Chemical, Midland, MI, USA) in a solvent of N, N-dimethylformamide (DMF) (Junsei, Kyoto, Japan) and tetrahydrofuran (THF) (Daejung, Republic of Korea) (1:1.5 (*v/v*)). Subsequently, the solution for hydrophilic fibers was obtained by dissolving PU and Pluronic F-127 (PF) (Sigma-Aldrich) to a concentration of 10% (*w/v*) and 10% (*w/v*) of solvent. Each mixture was stirred at 65 °C for 24 h until all the solutes were dissolved. To introduce double-faced properties on both sides, the PU membrane was synthesized first by electrospinning, and the PU-PF membrane was electrospun on top of it. For electrospinning, 10 mL of the PU solution was prepared in a 10 mL syringe with the use of a 23 G metal spinneret needle; this aliquot was then electrospun at a flow rate of 0.4 mL/h at an applied voltage set at 13.5 kV. After electrospinning the PU membrane, 5 mL of PU-PF solution was loaded and electrospun at the same conditions as PU. The metal-based collector was located 40 cm away from the spinneret needle and was rotated at 10 revolutions per minute for 60 h. The conditions of electrospinning are summarized in Table 1. The electrospun bilayer membrane was dried in an oven at 60 °C for 4 h and was exposed to ultraviolet (UV) light for 10 h to sterilize it.

### 2.4. Characterization of Electrospun Bilayer Membrane

A high-resolution scanning electron microscope (HR-SEM) (SUPRA55VP, Zeiss, Aalen, Germany) was employed to observe the surface morphology and thickness of the electrospun bilayer membrane. The bilayer membrane was prepared at a size of 10 mm (width) × 10 mm (depth) × 150 µm (height) for HR-SEM, and the PU and PU-PF surfaces were then imaged. To examine the wettability of each surface, the contact angle was measured using a contact angle meter (GSS, SurfaceTech, Ansan-si, Republic of Korea). The bilayer membrane was prepared on a flat holder and set to be perpendicular to a 27 G needle. Subsequently, single DI water droplets were released from a 3 mL syringe on the PU or PU-PF surfaces. To compare the wettability of the bilayer membrane, the folded membrane (trimmed to 20 × 90 mm) was attached to the slide glass (Marienfeld Superior, Lauda-Königshofen, Germany). The PU surface is located in the upper portion, and the PU-PF surface is located on the bottom, with both sides facing up, and vice versa, as shown in Appendix A. These prepared samples were tilted at 45°, and 300 µL of watercolor ink (red) was dropped onto the top of each sample. In addition, to determine the permeability, the bilayer membrane was prepared on a flat holder and set to be perpendicular to the 27 G needle with a 3 mL syringe. A fluorescent bead solution was made by mixing 1000 µL of DI water and 10 µL of red fluorescent beads (diameter 10 µm) (Micromod, Rostock, Germany). A single droplet of the fluorescent bead solution was released onto the PU or PU-PF surfaces. The adsorbed or permeated beads were observed using HR-SEM and fluorescent microscopy (Zeiss) on the PU or PU-PF surfaces. The image size was then converted to 0.651 pixels/µm ratio, and the trapped bead area was quantitatively analyzed and measured using Image J software (National Institutes of Health, Bethesda, Rockville, MD, USA). In addition, watercolor ink (red) was dropped on the boundary of the folded bilayer membrane to confirm the relative absorption on the two sides.

### 2.5. HA Hydrogel Arraying on Electrospun Bilayer Membrane

The HA hydrogel array was formed by photo-crosslinking the electrospun bilayer membrane and replicated the PDMS mold (Appendix A). After sterilization with autoclaving, the surfaces of the PDMS concave mold and electrospun bilayer membrane were treated with oxygen plasma at 60 W (Cute, Femto-Science, Hwaseong-si, Republic of Korea) for 30 s to facilitate filling with HA solution. Prelyophilized HA and 0.05% (*w/v*) 2-hydroxy-4′-(2-hydroxyethxy)-2-methylpropiophenone (TCI, Tokyo, Japan) were dissolved in Dulbecco’s phosphate-buffered saline (DPBS) (Welgene, Gyeongsan-si, Republic of Korea) to form the HA solution. The HA solution was applied to a concave PDMS mold and scratched to fill the inner parts of the pattern. Then, the residual solution was removed. The membrane (15 × 15 mm) was then used to cover the HA-filled concave pattern; the hydrophilic side of the membrane and pattern side of the PDMS mold faced each other. To achieve the photo-crosslinking of the HA solution on the membrane, the HA-filled PDMS mold was exposed to UV light (Omnicure S2000, Excelias Techonolhies Corp., Waltham, MA, USA) (wavelength: 360 nm, intensity: 10,000 mW/cm^2^) at 8 cm for 50 s [63]. Detachment of the PDMS mold from the membrane successfully led to the photo-crosslinking of the HA hydrogel array on the bilayer membrane. Additionally, the utility of the HA hydrogel array on the hydrophilic layer was compared with those on the hydrophobic layer. The 2.5, 5.0, and 10% (*w/v*) HA solutions were used for the HA hydrogel array, and the perfect single array was counted depending on the HA concentration. To confirm the HA hydrogel arraying on the hydrophilic surface, the 10 µm fluorescent beads were added to the HA solution to observe the array formation using fluorescent and confocal microscopy (LSM880 with Airyscan, Zeiss).

### 2.6. Diffusion Test

To study the diffusion profile from the immobilized substance HA hydrogel array on the electrospun bilayer membrane, a previously reported protocol was employed [64]. Briefly, fluorescein–isothiocyanate-labeled bovine serum albumin (FITC-BSA) was encapsulated in 2.5, 5, and 10% (*w/v*) solutions, and was refreshed with 2 mL of DI water daily for 6 days. Each time the DI water was refreshed, the fluorescence intensities of the same spots on all HA hydrogel arrays on the electrospun bilayer membranes were recorded. To investigate the role of the membranes for mass transport as a substrate at different wettability values, a 3D environment was emulated using a lab-made rectangular acrylic cube with a size of 25 mm (width) × 20 mm (depth) × 30 mm (height). The assembled acrylic cube was filled with DI water through its opened upper side. The watercolor ink (blue) was mixed in the HA hydrogel array, immobilized on the membrane, and placed on the top of the acrylic cube toward the inner parts of DI water, or in the opposite direction (Appendix A). The diffusing profile from the HA hydrogel array was observed and tracked with a digital camera (Canon, Tokyo, Japan). The approximation equation for diffusion time is as follows:(1)$$t\approx \frac{x^{2}}{2D}$$
where *D* denotes the diffusion coefficient of an ink, x the mean distance traveled by 5-FU via diffusion, and *t* the elapsed time since diffusion began [65]. Finally, to simulate the delivery of 5-FU from the HA hydrogel arrays, a 2% (*v/v*) diluted FITC-BSA solution was mixed with 2, 5, and 10.0% (*w/v*) HA hydrogel solutions to prepare the three different membranes. Then, the fabricated membranes were used to cover the porcine tissue cut to a size of 20 mm (width) × 20 mm (depth) × 10 mm (height). The samples were stored in a glass storage container at 36.5 °C and 40% humidity; after 12 h, the fluorescence intensities were measured using a fluorescence microscope.

### 2.7. 5-FU Immobilization and Release with YD-10B Cells

To estimate the rate of tumor cell death due to the controlled release of the anticancer drug, YD-10B cells derived from human oral squamous cancer were cultured with 5-FU at different concentrations, 1, 2 and 3% (*v/v*), for a fixed 10% (*w/v*) HA hydrogel array on the electrospun bilayer membrane. The membrane was prepared at a size of 20 × 20 mm and was fixed at the bottom of the cell culture dish with the use of PDMS blocks as supporters to prevent floating in the culture media. YD-10B cells (100 µL, cell density of 1.0 × 10^6^ cells/mL) were seeded on the HA-hydrogel-arrayed membrane that contained 5-FU [66,67,68]. As a control group, YD-10B cells without 5-FU were seeded on the array. In addition, the workability and extended release of the HA array was compared with cultured YD-10B cells on a culture dish without the HA-arrayed bilayer membrane under the same concentrations of 5-FU. The cells were cultured in culture media (RPMI (RPMI 1640 medium, HEPES)) (Gibco, San Diego, CA, USA) with 10% fetal bovine serum (FBS) (Gibco) and 1% penicillin/streptomycin (Gibco) in an incubator (MCO-18AIC, Sanyo, Japan) at 5% CO_2_ and 37.5 °C. After 3 days, the live/dead assay (LIVE/DEAD^®^, viability assay kit (Invitrogen, Waltham, MA, USA)) was conducted to evaluate the anticancer performance. The solution for the live/dead assay was mixed with 20 µL of ethidium homodimer-1 (EthD-1) and 5 µL of calcein acetoxymethyl (calcein-AM) in 10 mL of DPBS in the dark room. The culture media were removed using a micropipette, and the residual media were washed twice using DPBS. The staining solution was added to the HA hydrogel-arrayed membrane and incubated for 40 min. The viability of the YD-10B cells was evaluated using a fluorescent microscope. The cytoplasm of live cells was stained with calcein-AM (green fluorescence, excitation wavelength: 488 nm/emission wavelength: 515 nm), whereas the DNA of dead cells was stained with EthD-1 as red fluorescence (excitation: 570 nm/emission: 602 nm).

## 3. Results and Discussion

### 3.1. Electrospun Bilayer Membrane

The electrospinning-based electrospun bilayer membrane is shown in Figure 1. Both the PU and the PU-PF surfaces of the membrane have well-defined straight fibers that are randomly stacked and form numerous pores (Figure 1a,b). The cross-section of the bilayer membrane is shown in Figure 1c; the thickness of the PU and PU-PF layers were measured to be equal to 50 and 110 µm, respectively. As shown in Figure 1c, different surfaces had different wettability values. The measured contact angle on the hydrophobic surface was 96.1°; conversely, on the hydrophilic surface, the contact angle was 13.58°. We assumed the properties of PF with the hydrophilic polyethylene glycol chains reported previously [69,70,71,72,73,74,75,76]. The properties of the bilayer membrane were verified and quantified by the wettability tests (Figure 2). When the hydrophobic PU surface was located at the top, the watercolor ink (red) flowed downward along the slope without absorption but was absorbed into the hydrophilic PU-PF surface at the bottom (Figure 2a, Appendix A). Conversely, watercolor ink (red) was rapidly absorbed and did not flow downward to the bottom when the hydrophilic PU-PF surface was located at the top (Figure 2b, Appendix A). The reason for this is that PU is generally known as a water-repelling material because of the presence of an -NO_2_ functional group, but this has been changed to a water-attracting property by the polypropylene oxide functional group in PU-PF (Appendix A). In addition, a solution that contained fluorescent beads was dropped onto both surfaces of the membrane to observe the number of beads remaining (Figure 3). The two types of folded membrane were tilted by 45°, and the bead solution was dropped at the same position to compare the number of residual beads on the membrane. The number of residual beads on the PU-PF surface outnumbered that on the hydrophobic PU surface as the solution was absorbed in the hydrophilic PU-PF surface (Figure 3a,b). In the quantitative analysis of the trapped beads within areas equal to 2.3 mm^2^, these occupied an area of 0.017 mm^2^ on the hydrophobic surface and 0.265 mm^2^ on the hydrophilic surface (Figure 3c,d). The area of the beads trapped on the hydrophilic surface was calculated to be 3.7 times higher than that on the hydrophobic surface. The surface energy, which determines the wettability of electrospun bilayer membranes, was evaluated by the contact angle, absorptiveness, and dispersive influence tests. In the PU, only the electrospun surface was tested, and the measured contact angle was high because the PU had low surface energy and the dropped DI water had high surface tension. However, the contact angle was lowered by the increased surface energy due to the hydrophilic polar group, which was introduced onto the surface by Pluronic F-127. In this regard, it can also be explained why the injected fluorescent beads were absorbed together with the solution on the hydrophilic surface. These results indicate the successful physical properties of a membrane that can selectively permeate or repel materials owing to the wettability of individual surfaces. For qualitative analyses, the spectra of both sides of the membrane, obtained with the use of the FT-IR spectrophotometer, are shown in Appendix A. The FT-IR spectra for both the PU and PU-PF membranes were similar to those in previously published reports [77,78]. Appendix A shows that the PU membrane spectrum had sharp bends at 1728 and 1701 cm^−1^, which were assigned to C=O stretching. The peaks at 3325 and 1103 cm^−1^ were related to –NH group and C–O–C stretching, respectively. In the PU-PF spectra, the C=O 1728–1701 cm^−1^ peaks were attributed to the PU membrane. However, according to the overlapped –CH_2_ (SP^2^) peaks at 2854–2939 cm^−1^ and the –CH_2_ (SP^3^) peak at 1342 cm^−1^, the –OH group broadly overlapped at the N–H group (3325 cm^−1^). Therefore, the peak at 1102 cm^−1^ for the PU-PF side was approximately 1.5 times higher and may have been derived from the C–O–C bond of PF used for hydrophilization.

### 3.2. HA Hydrogel Array

The crosslinked HA hydrogel array on the hydrophilic surface of the bilayer membrane, based on the use of a 15% (*w/v*) HA solution, is shown in Figure 4. In this process, UV light must reach the HA solution on the prepared membrane through the transparent PDMS mold. Additionally, the crosslinked HA hydrogel should be separated from the PDMS mold while maintaining attachment with the membrane. On the hydrophilic PU-PF surface, the HA solution was rapidly absorbed into the membrane and was crosslinked with a high adherence so that it could be easily separated from the PDMS mold. For this reason, the integrated hydrogel array has a hemispherical shape with a diameter equal to 700 µm, the same size as the designed and manufactured PDMS mold (Figure 4a). Therefore, the mixed fluorescent beads were observed only in the crosslinked HA hydrogel array and not in other parts of the bilayer membrane (Figure 4b). As shown in Appendix A, it is believed that arranging HA solutions with different concentrations on the hydrophilic PU layer was difficult due to the deficient hydrophilic functional group. Although the shape of some single arrays was formed adequately due to the relatively high viscosity in the 10% (*w/v*) HA solution, this has a low yield rate of 30%. In addition, the production rate of HA hydrogel arrays on the hydrophilic layer was increased depending on higher HA concentrations. However, the 2.5 and 5.0% (*w/v*) HA solutions still have insufficient production rates of 22.2 and 61.1%, respectively (Figure 4c). Therefore, this study attempted the HA hydrogel array on the hydrophilic layer, and drug release was induced by controlling degradation according to concentration. Specifically, the lyophilized HA hydrogel was confirmed by using the FT-IR spectra (Appendix A). The main functional groups of synthesized HA were identified in previous research studies [79,80]. Briefly, the FT-IR spectrum shows an intense signal at 3277 cm^−1^ assigned to the N–H or O–H stretching vibrations. These broad peaks are assigned to the presence of alcohols–phenols, amines–amides, and carboxylic acids. The moderate peak at 2890 cm^−1^ may be due to the C–H stretching of the alkyl chain and indicates the presence of aromatic C–H bonds observed in the 3200 to 3000 cm^−1^ regions; these can be overlapped by broad N–H and O–H peaks. Additionally, the SP^3^ C–H peaks at 1405 and 1376 cm^−1^ (methyl and methylene group) and vibrations of alkoxy C–O bonds in ether, phenol, and alcohol at 1028 cm^−1^ support the presence of hydrophilic functional groups, such as O–H or –COOH. In terms of the cross-linking of the HA hydrogel on the membrane based on wettability, the hydrophobic environment is not suitable because the pre-crosslinked HA hydrogel is repelled from the PU surface. The reason for this is that PU is a water-repelling material due to the presence of the –NO_2_ functional group. In the preliminary experiment, we tried to fabricate a patch using an electrospun PU membrane; however, an HA array did not easily build a structure and was separated from the membrane even after crosslinking because of the high water repellency of the electrospun PU surface. In contrast, in the pre-crosslinked HA hydrogel, array formation was observed on the highly hydrophilic PU-PF membrane even at low HA hydrogel concentrations. Therefore, we believe that these hydrophilic functional groups (–OH, –COOH, and –NH) in the PU-PF membrane and HA hydrogel can precisely control the hemispherical arraying for drug delivery. Furthermore, it is suggested that the proposed process can easily and successfully inject and control the target materials within the HA hydrogel array.

### 3.3. Diffusion Aspects

To demonstrate the encapsulation and release potential of materials based on HA hydrogel arrays, FITC-BSA was added to the HA solution and crosslinked to membranes in the array. Figure 5 shows the intensity profile of the fluorescence intensity changes of the HA hydrogel over time after the fluorescence intensity value inside the pattern was specified to be 1.00 based on the first measured intensity value after the formation of the array. The relative fluorescence intensity values of the array after 10 h in each condition were reduced to approximately 0.44, 0.53, and 0.65 in 2.5, 5.0, and 10.0% (*w/v*) HA concentrations, respectively. The calculated release rate was approximately 1.2 and 1.5 times higher in the 2.5% (*w/v*) compared with the 5 and 10.0% (*w/v*) HA hydrogels. On day 5 (7200 min), the amount of FITC-BSA released from the HA hydrogels became similar, and the measured fluorescence intensity values were 0.15, 0.16, and 0.17 for the 2.5, 5.0, and 10.0% (*w/v*) HA hydrogel concentrations, respectively. This means that the release control of the target substance can be controlled by the concentration and array of the HA hydrogel.

In terms of the role of the bilayer membrane (which possesses both hydrophobic and hydrophilic features), a 3D environment was constructed, and the diffusion phenomenon of the immobilized material was observed as being different depending on the wettability. Given that the hydrophobic surface of the electrospun membrane could not be arrayed owing to the low surface tension of the membrane, only the hydrophilic surface was arrayed by mixing the same concentration of watercolor ink (blue) and HA hydrogel. As shown in Figure 6a–d, when the unpatterned hydrophobic surface inside the water-filled cube was in contact with water in a static state, the watercolor ink slowly diffused toward the water from the HA hydrogel array after approximately 5 min. The diffusion coefficient values measured under each condition were 0.00286 cm^2^/s for the hydrophobic surface and 0.02500 cm^2^/s for the hydrophilic surface, which is approximately 8.74 times higher in the hydrophilic surface. In addition, it was observed that the hydrophobic membrane was pulled owing to the influence of the surface tension of the filled water. In contrast, as shown in Figure 6e–h, when a membrane arrayed on a hydrophilic surface contacted water, the watercolor ink diffused into the water at a high rate throughout the HA hydrogel array. Owing to the high surface tension of the membrane on the hydrophilic surface in contact with water, the membrane was not contacted by water and could maintain its original shape. These results demonstrate that the hydrophobic surface of the bilayer membrane acts to inhibit or impede the passage and transportation of substances, whereas the hydrophilic surface can facilitate both the immobilization and transportation of substances.

Figure 7 shows the simulated time-dependent FITC-BSA concentration distributions according to the three different concentrations of the HA hydrogel arrays for 12 h. For the HA hydrogel array at a concentration of 2.5% (*w/v*), most of the HA hydrogel convex structures were not morphologically observed on the membrane (Figure 7a,d), and FITC-BSA molecules were expressed but widely spread throughout the porcine tissue (Figure 7g). For the 5% (*w/v*) HA hydrogel concentration, the HA array remained on the membrane but with an imperfect convex structure, and the fluorescence was measured to be relatively reduced in intensity (Figure 7b,e). FITC-BSA partially diffused from the epidermal layer to the dermal layer of the porcine tissue (Figure 7h). As expected, for the 10% (*w/v*) concentration of the HA hydrogel, FITC-BSA was almost exclusively expressed within the epidermal layer of the porcine tissue (Figure 7i), and high-intensity fluorescence expression was observed in the HA hydrogel array on the bilayer membrane (Figure 7c,f). Although the porcine tissue was spoiled due to the harsh experimental environmental conditions and could only be observed for 12 h, under the same diffusion conditions for the low-concentration HA hydrogel, most of the FITC-BSA molecules were released within a short time, and the diffusive transfer seemed to be rapid. Conversely, when the concentration of the HA hydrogel was relatively high, the diffusion transfer of FITC-BSA was not rapid from the HA hydrogel array to the porcine tissue. These results indicate that 5-FU, rather than FITC-BSA, can be locally immobilized within the HA hydrogel array and released at differently controlled rates.

### 3.4. Effect of 5-FU Local Release

Human oral squamous cell carcinoma cells (YD-10B) were cultured for 3 days on three different HA hydrogel arrays containing 5-FU at concentrations of 1, 2, and 3% (*v/v*), and on another HA array without 5-FU immobilization as a control. The gray circles shown in the figure indicate the portion of the HA array on the membrane. Figure 8a shows a survival rate of over 90% after 3 days in the absence of 5-FU. In contrast, only a few cells survived and were adhered to the membrane in all of the experimental conditions that included 5-FU, as shown in Figure 8b–d. Only a small number of cells were identified using the Live/Dead kit because most of the cells had been damaged or died even prior to membrane adhesion because of the release of 5-FU from the HA hydrogel arrays. On the second day, at a fixed at a concentration of 1% (*v/v*) 5-FU, the survival rate was about 60% even among the small number of adherent cells. When the concentration of 5-FU was higher, at 2% (*v/v*) or 3% (*v/v*), the measured survival rates were decreased to less than 40% and 10%, respectively. On the third day, for the 1% (*v/v*) 5-FU, the survival rate was about 20%; for the 2% and 3% (*v/v*) 5-FU, the cell viability was less than about 10%. These results were compared with an experiment (Appendix A) where YD-10B cells were cultured in normal cell media in 6-well plates without the HA arrays and using the same concentrations of 5-FU. In these environments, the cells started being affected by 5-FU on the third day rather than on the second day, revealing reduced viability of the YD-10B cells. However, when 5-FU is diluted in the culture media, the initial adhesion rate and survival rate of cancer cells do not seem to be significantly altered. This could be because the 5-FU molecules are evenly diffused within the media and are not focused in one area as they are when using membrane delivery. Our results show that 5-FU immobilized in HA hydrogel arrays affects cells even at the very early stages of cell culture, significantly reducing YD-10B cell adhesion and proliferation. These results show that when 5-FU is immobilized within HA hydrogel array on the bilayer membrane as a drug patch, drug concentration control and sustained release could be possible and applicable to target topical tumor sites.

## 4. Conclusions

In this study, locally controlled release of 5-FU attached to a specific site was achieved using a biocompatible HA hydrogel array on an electrospun bilayer membrane to control the side effects of the drug burst at an early stage. The HA hydrogel array was designed for topical fixation and release of 5-FU. HA and MA were used to synthesize the hydrogel used as carriers to protect and deliver the 5-FU. The drug was loaded onto a 3D-reticulated electrospun bilayer membrane by crosslinking a solution containing the 5-FU. In terms of the electrospun bilayer membrane, the hydrophilic layer played a role in the formation of the HA array and cell adhesion, and a hydrophobic layer serving as a three-dimensional structure in the form of a porous scaffold was proposed. YD-10B cells were seeded onto the 5-FU drug-encapsulated HA hydrogel array and were cultured for 3 days as an experimental group. Under the conditions where the 5-FU was not fixed, cells showed normal adhesion and a high proliferation rate. On the other hand, under the conditions of immobilization, 5-FU was locally released from the HA hydrogel array, most of the cells died without sticking, and only a few were attached to the membrane. The proposed method is expected to be applicable to the field of patch-type chemotherapy or tissue engineering that can simply fix and release various types of anti-cancer drugs or therapeutic agents.

## Figures and Tables

**Figure 1 bioengineering-09-00742-f001:**
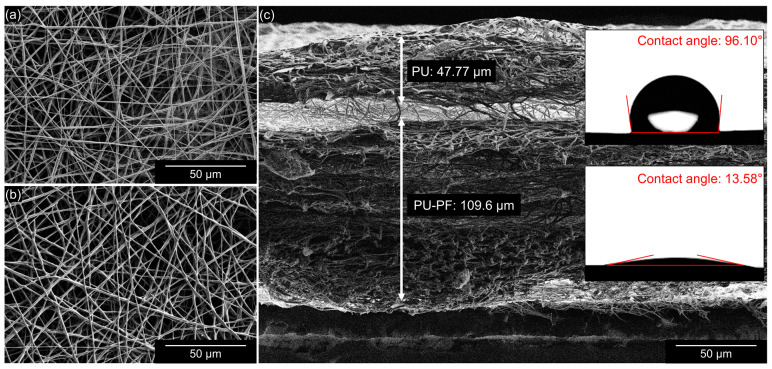
Scanning electron microscopy (SEM) images of (**a**) hydrophobic (PU) side, (**b**) hydrophilic (PU-PF) side, and (**c**) cross section of electrospun bilayer membrane.

**Figure 2 bioengineering-09-00742-f002:**
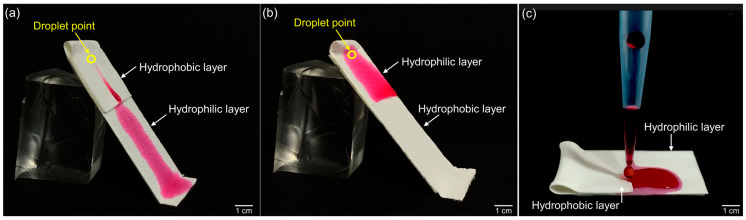
Results of wettability test on the tilted bilayer membrane. (**a**) The absorbed watercolor ink (red) on hydrophilic surface at lower area and (**b**) upper area, and (**c**) the aspects of repelling and attracting water on the bilayer membrane.

**Figure 3 bioengineering-09-00742-f003:**
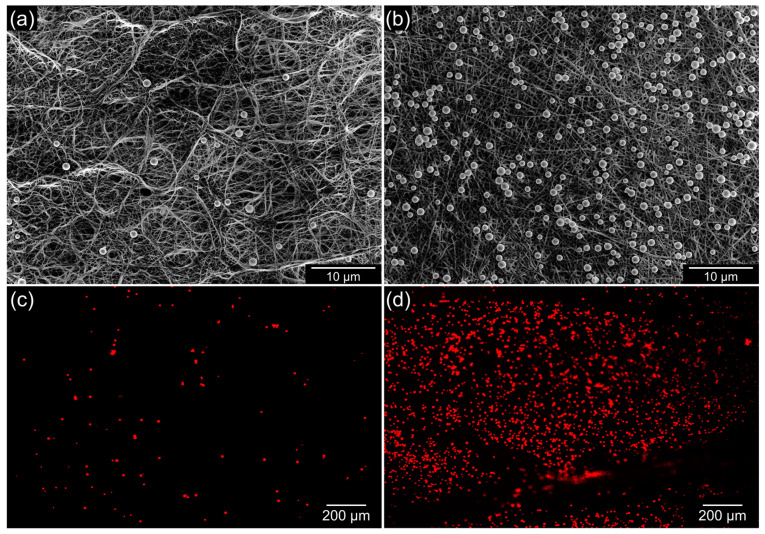
Results of trapped beads on each side of bilayer membrane. (**a**,**b**) SEM images and (**c**,**d**) fluorescent images of residual beads.

**Figure 4 bioengineering-09-00742-f004:**
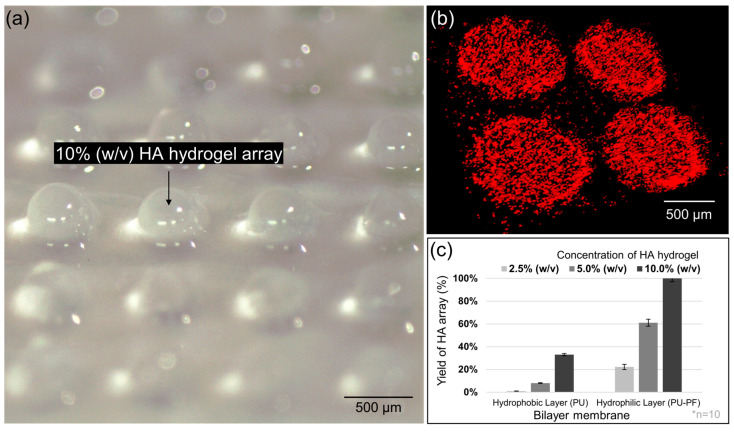
The results of the cross-linked HA hydrogel array. (**a**) HA hydrogel array on the hydrophilic surface of the electrospun bilayer membrane. (**b**) The images of encapsulated 10 µm fluorescent beads in the HA hydrogel array. (**c**) Comparison of yield of HA array on PU (hydrophobic) and PU-PF (hydrophilic) layers.

**Figure 5 bioengineering-09-00742-f005:**
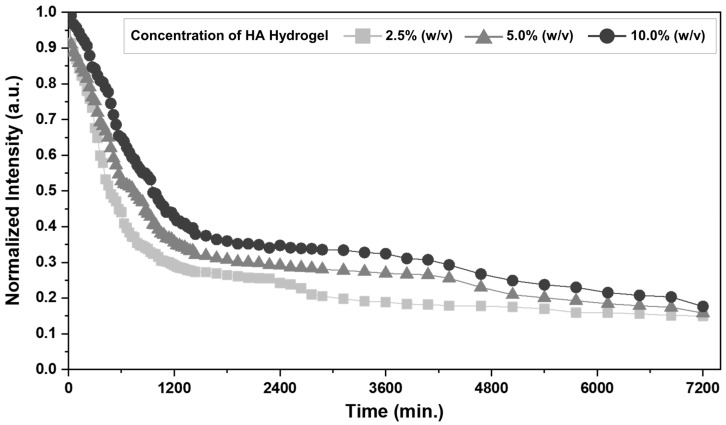
Release profiles of FITC-BSA according to HA hydrogel concentration (2.5% (*w/v*), 5% (*w/v*), and 10.0% (*w/v*)).

**Figure 6 bioengineering-09-00742-f006:**
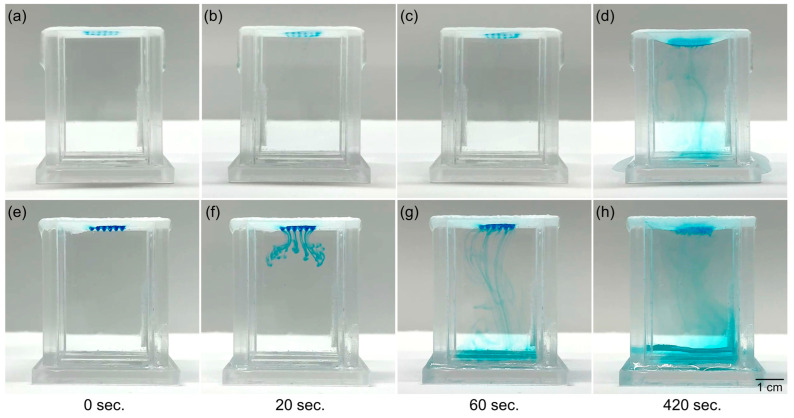
The role of electrospun bilayer membranes for diffusion profiles as a substrate at different wettability values. (**a**–**d**) The tracked images of HA hydrogel array mixed with watercolor ink (blue) toward the upside and (**e**–**h**) in the opposite direction.

**Figure 7 bioengineering-09-00742-f007:**
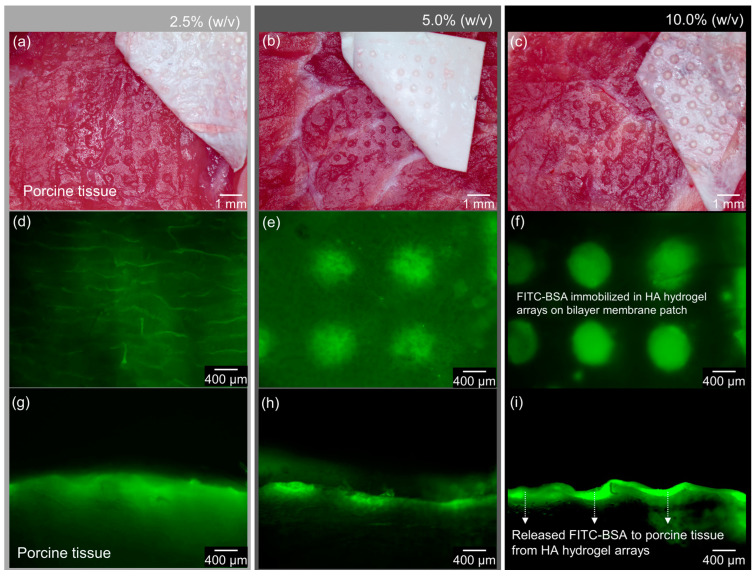
Images of the porcine tissue with the bilayer membrane containing the FITC-BSA-immobilized HA hydrogel array. (**a**) HA hydrogel array at a concentration of 2.5% (*w/v*), (**b**) HA hydrogel array at a concentration of 5.5% (*w/v*), (**c**) HA hydrogel array at a concentration of 10% (*w/v*), (**d**–**f**) different expression of fluorescence in HA hydrogel arrays on bilayer membranes at concentrations of 2.5, 5, and 10% (*w/v*), and (**g**–**i**) diffused fluorescence from HA hydrogel arrays on bilayer membranes to porcine tissue.

**Figure 8 bioengineering-09-00742-f008:**
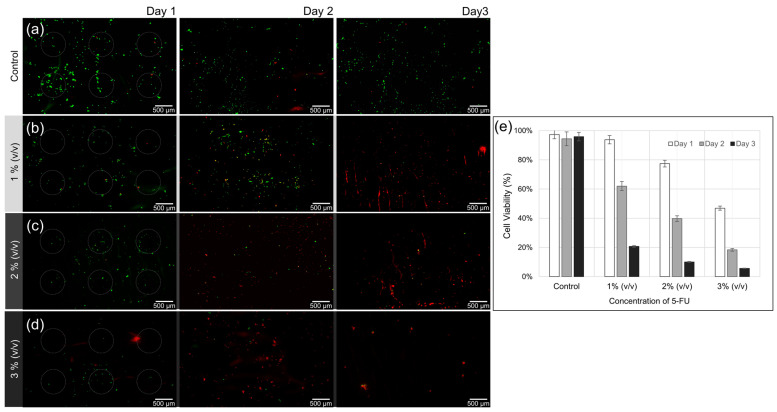
Live/dead assay images of YD-10B cells on a bilayer membrane containing HA hydrogel array. (**a**) Control group, (**b**) concentration of 1% (*v/v*) 5-FU (**c**) concentration of 2% (*v/v*) 5-FU, (**d**) concentration of 5% (*v/v*) 5-FU, and (**e**) viability of YD-10B cells on a bilayer membrane containing HA hydrogel array.

**Table 1 bioengineering-09-00742-t001:** Condition of electrospinning for bilayer membrane.

Polymer	Solvent	Concentration	Total Volume
Polyurethane (PU)	Dimethylformamide:Tetrahydrofuran = 1:1.5 (*v/v*)	10% (*w/v*)	10 mL
PU-Pluronic F-127 (PU-PF)		PU: 10% (*w/v*) PU-PF: 10% (*w/v*)	5 mL
**Voltage**	**Tip-to-collector distance**	**Flow rate**	**Needle gauge**
13.5 kV	40 cm	0.4 mL/h	23 G

## Data Availability

The data presented in this study are available upon request from the corresponding author.

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
