# Peer review of "5-Fluorouracil-Immobilized Hyaluronic Acid Hydrogel Arrays on an Electrospun Bilayer Membrane as a Drug Patch"

_bioengineering, 2022, doi:10.3390/bioengineering9120742_

Round 1
Reviewer 1 Report
HA is a commonly used drug delivery system that has been widely used, the authors showed a new method to delivery drug in their paper. Before it can be accepted for publication, there authors should answer the reviewer's comments as follows:
Figure 2, it is not ideal to put the drop point at the same level, because in the (a) panel, there was a large amount of solution on the bottom edge of the hydrophobic surface; in the (b)panel, since it is hydrophilic, there was almost no solution on the edge of the bottom; thus made the comparison not scientific, the authors should put similar amount of solution just near the bottom of the edge in the hydrophilic surface like the amount of the hydrophobic surface.
Figure 8 and 9: the authors need another control of 5-FU, since you are not compare the difference of 5-FU, but compare the new drug delivery system. comparing with or without 5-FU can not support your point of view.
Author Response
"Please see the attachment"
Dear review
We appreciate your comments in which you requested revision reports to the manuscript entitled:
(before) 5-Fluorouracil-Immobilized Hyaluronic Acid Hydrogel Arrays on Electrospun Bilayer Membrane as Drug Carrier
(After) 5-Fluorouracil-Immobilized Hyaluronic Acid Hydrogel Arrays on Electrospun Bilayer Membrane as Drug Patch (manuscript number: Bioengineering-2026868).
We have thoroughly reviewed the manuscript and have made changes as per reviewers’ suggestions. We believe that the revised manuscript would be more improved. Below are our specific points by point comments for each reviewer and attached please find a copy of the paper with highlighted modifications.
We believe that after addressing the reviewer’s questions and concerns the revised manuscript is much stronger. We thank the reviewers for their constructive feedback and look forward to hearing from you regarding the status of our manuscript for publication in Bioengineering Journal.
Thank you.
Best regards

Reviewer 2 Report
The authors developed a hyaluronic acid (HA) hydrogel array with an electrospun bilayer membrane for the topical release of 5-Fluorouracil (5-FU) to the cancer site. The complex and advanced processing technologies (3D printing and electrospinning) of 5-fluorouracil-immobilized hyaluronic acid hydrogel array is innovative. However, the effectiveness of anticancer and drug release of such a hydrogel array and its mechanism is not sufficiently investigated in vitro and in vivo to support the authors’ claim. Overall, such a confused experimental design may not meet the criteria for publication in Bioengineering. The following issues should be considered in the revisions:
1. In the abstract and introduction, the author spends considerable space to explain the anticancer effect and significance of 5-FU, but in the subsequent experimental verification, there is a lack of corresponding biological characterizations (cell and tissue) in vivo and in vitro to demonstrate it’s drug release and anticancer effect.
2. In the abstract and introduction, the absence of design strategy and discussion for this integrated and complex processing technologies and structure makes the innovation and significance of this work haven’t been emphasized. It is suggested that the authors should supplement the research background, recent progress, and scientific problems on drug carrier or hydrogel-based materials to indicate the significance and prospect of this work. Some of the recent reports about hydrogels can be discussed in this work (for example, Nature Communications, 2020, 11, 1604; Chemical Engineering Journal, 2022, 442, 136284).
3. The conclusion of cell’s survival rate (98% in 5 days) in the abstract and subsequent results and discussions are not corresponding. We cannot understand how the authors obtain the results of survival rate without corresponding tests, such as the CCK-8 or MTT staining. My suggestion is that the author should supplement corresponding experiment.
4. Line 300-303 on page 8, the author claims that “Therefore, we believe that these hydrophilic functional groups (-OH, -COOH, and -NH) of the PU-PF membrane and HA hydrogel can precisely control the hemispherical arraying for drug delivery. Furthermore, it is suggested that the proposed process can easily and successfully inject and control the target materials within the HA hydrogel array.” without the explanation of controlled conditions and experimental demonstration for drug delivery. I advise the author to verify the possibility and effectiveness of drug delivery, rather than draw conclusions indirectly through the existence of hydrophilic functional groups.
5. In page 11, the illustrations of Figure 8 and 9 are confused, and the experimental phenomena and condition cannot be explained clearly, especially the meaning of the red circle. Additionally, the experimental phenomena can’t prove that “numerous cells aggregated and grew around the array”, due to several cells only aggregating around the right and rarely attaching around other arrays (Figure 8a).
6. The title of the article needs to be changed.
Author Response

(The authors gave the same response as above.)

Round 2
Reviewer 1 Report
The revision has greatly improved, and the reviewer's questions have been answered.
Author Response
"Attachment is same contents as below"
In Round 1, we received a reply from a professional reviewer as presented below.
[The revision has greatly improved, and the reviewer's questions have been answered]
Once again, we would like to thank you again for your very valuable comments to improve the quality of the submitted manuscript.
Based on the recommendation of the academic editor, after receiving English proofreading through a professional institution, we submitted the revised manuscript with a proofreading certificate
We appreciate your response.
Best regards,
Kwang-Ho Lee

Reviewer 2 Report
I would like to recommend this paper for publication in Bioengineering.
Author Response
"Attachment has same contents as below"
In Round 1, we received a reply from a professional reviewer as presented below.
Once again, we would like to thank you again for your very valuable comments to improve the quality of the submitted manuscript.
[I would like to recommend this paper for publication in Bioengineering.]
Based on the recommendation of the academic editor, after receiving English proofreading through a professional institution, we submitted the revised manuscript with a proofreading certificate
We appreciate your response.
Best regards,
Kwang-Ho Lee
